# Factors associated with quality of life of chronic spontaneous urticaria patients in a Vietnamese population

Nguyen Thi Tra My[1,2], Le Huyen My[3], Vu Nguyet Minh[1,3], Nguyen Thi Ha Vinh[1,3], Mai Ba Hoang Anh[2], Doanh Le Huu[1,3]*

1 Department of Dermatology, Hanoi Medical University, Hanoi, Vietnam, 2 Department of Dermatology, Hue University of Medicine and Pharmacy, Hue University, Hue, Vietnam, 3 National Dermatology and Venereology Hospital, Hanoi, Vietnam

* lehuudoanh@gmail.com

**Data Availability Statement:** Data cannot be shared publicly because of the restriction regulations of IRB in Hanoi Medical University for protecting personal data and information. Data are

## Abstract

### Objective

Chronic spontaneous urticaria (CSU) is a challenging condition that significantly impacts the affected patients. This study aimed to evaluate the quality of life (QoL) among patients with CSU in Vietnam and identify factors associated with QoL.

### Methods

A cross-sectional study was conducted at the Vietnam National Dermatology and Venereology Hospital from June 2023 to March 2024. A total of 358 CSU patients aged 16 years or older were recruited. Data were collected using a structured questionnaire covering demographic, clinical, and laboratory characteristics. The Chronic Urticaria Quality of Life Questionnaire (CU-Q$_2$oL) and the Weekly Urticaria Activity Score (UAS7) were utilized to assess QoL and disease severity. Multivariate Tobit regression models were performed.

### Results

The CU-Q$_2$oL total score had a mean of 48.67 (SD = 16.90) and a median of 46 (IQR = 35–59). The scores for individual CU-QoL subscales were as follows: pruritus (5.42±2.02), swelling (2.86±1.54), life activities (13.89±6.00), sleep problems (11.12±4.96), limits (6.52±2.66), and looks (8.85±4.09). Higher UAS7 scores were associated with lower QoL, and angioedema in the eyes and lips were associated with increased swelling domain and poorer overall QoL. Longer disease duration was associated with higher pruritus scores, while a history of allergy was related to poorer total QoL, sleep, and looks. Severe itching further degraded sleep quality. Positive Autologous Serum Skin Test (ASST) was correlated with lower overall QoL, particularly in swelling and limits domains. Positive Basophil Histamine Release Assay (BHRA) status was linked to poorer sleep quality domain.

available from the IRB of Hanoi Medical University with reasonable reasons (contact: daihocyhn@hmu.edu.vn).

**Funding:** The author(s) received no specific funding for this work.

**Competing interests:** The authors have declared that no competing interests exist.

## Conclusion

CSU significantly impairs the QoL of patients, affecting physical, emotional, and social dimensions. Regular QoL assessments should be integrated into clinical practice to ensure comprehensive and patient-centered treatment strategies.

## Introduction

Chronic spontaneous urticaria (CSU) is a challenging condition that significantly impacts the quality of life (QoL) of affected individuals [1]. This condition, characterized by the development of wheals and/or angioedema for more than six weeks without identifiable triggers, has an estimated point prevalence of around 0.5% to 1% [2]. CSU has significant detrimental effects on QoL, with sleep deprivation and psychiatric comorbidities being frequent [3]. Until the approval of omalizumab in early 2014, the primary symptomatic treatment for CSU was H1-antihistamines. Despite this, more than 60% of patients continued to experience symptoms even when treated with second-generation H1-antihistamines at the standard dose, and over one-third of these patients still did not respond to increased doses [4].

The burden of CSU is substantial, as it adversely affects the patient's QoL, disturbs daily activities and sleep patterns due to pruritus, and causes emotional disorders and social isolation [5]. Assessing QoL in CSU patients is critical for comprehensive disease management [6–8]. QoL measures provide insights into how the condition affects patients' daily activities, emotional states, and overall life satisfaction [9]. In literature, the impact of CSU on QoL is further emphasized by the fact that the disease seriously compromises the QoL of patients, with debilitating and uncomfortable symptoms that may last for years [6–8]. While there are several high-quality general and dermatological QoL questionnaires available, none specifically cater to the unique QoL concerns of CSU patients. The Chronic Urticaria Quality-of-Life Questionnaire (CU-$Q_2$oL), a tool designed specifically for CU patients has been introduced that is expected to significantly benefit both clinical research and everyday patient care by offering a more precise evaluation of the QoL impact in this particular group [10, 11].

In many countries, the QoL of CSU patients are often overlooked aspects of clinical practice [9, 12, 13]. Existing studies have highlighted that the severity of itching, frequency of urticaria episodes, and presence of angioedema are significant determinants of reduced QoL in CSU patients [tltk]. Despite the global recognition of the importance of QoL assessments in CSU management [9], more localised evidence is needed in the Vietnamese context. The unique socio-cultural, economic, and healthcare dynamics in Vietnam necessitate specific studies to understand how CSU affects Vietnamese patients. Gathering local evidence will enable healthcare providers to develop culturally appropriate management strategies and policies that address the unique needs of this population. This study aims to evaluate the QoL in patients with CSU in Vietnam. By identifying the key factors influencing these aspects, the research seeks to provide evidence-based recommendations to enhance patient care and improve overall disease management outcomes in the Vietnamese setting.

## Materials and method

### Study settings and patients

A cross-sectional, single-center study involving CSU patients aged 16 years or older was conducted at the Urticaria Clinic at the Vietnam National Dermatology and Venereology Hospital

from 1 June 2023 to 31 March 2024. Inclusion criteria required a diagnosis of CSU based on EAACI guidelines, characterized by the spontaneous appearance of wheals (hives) or angioedema for six weeks or more without any identifiable trigger factors. Wheals were defined as red or pink raised areas lasting less than 24 hours, while angioedema involved swelling of subcutaneous and submucosal tissues lasting up to 72 hours. Symptoms had to occur daily or almost daily for more than six weeks. Prior to performing the Autologous Serum Skin Test (ASST), patients who were using NSAIDs + antihistamines were required to discontinue them at least one week; and patients who were using or systemic corticoid were required to discontinue them at least one month. Additionally, patients exhibiting abnormal clinical, or laboratory signs were further screened to identify underlying causes. For instance, patients with fever or arthralgia were screened for infections, while those with eosinophilia underwent screening for parasitic infections. Any cases where a specific cause for the urticaria was identified were excluded from the study. This process ensured that the study focused only on patients with true CSU, free from external aggravating factors. A total of 388 patients were recruited, and data of 358 patients were used for analysis (response rate: 92.2%). Written informed consent was obtained from all participants. The study was approved by the Institutional Review Board of Hanoi Medical University (number 865/GCN-HDDDDNCYSH-DHYHN) and adhered to Good Clinical Practice and local regulations.

## Data collection

Information on CSU patients was collected using a structured medical record. The data collected included demographic information (age and gender), clinical characteristics, laboratory characteristics, and QoL. Investigators were dermatologists trained to examine and collect information consistently.

Clinical information collected included history of allergy, history of autoimmune disease, history of urticaria (no/acute/chronic), presence of comorbid chronic inducible urticaria (CIndU, which were test by challange tests for dermographysm, cold urticaria, cholinergic urticaria; the diagnosis of delayed pressure urticaria is made through physical examination, imaging, and medical history), presence of angioedema (eye/lip/limbs/throat/none), current itching severity (mild/moderate/severe), number of days with urticaria per week, duration of wheals, and history of CSU treatment. Current itching severity was assessed with the following levels: mild (present but not annoying or troublesome), moderate (troublesome but does not interfere with normal daily activity or sleep), and severe (severe itch, which is sufficiently troublesome to interfere with normal daily activity or sleep). These levels were consistent with the Weekly Urticaria Activity Score (UAS7) scale [14].

Moreover, the entire Weekly Urticaria Activity Score (UAS7) was also employed to measure the severity of CSU over a seven-day period [14]. It combines daily scores for the number of wheals (hives) and the intensity of itching, with each parameter scored on a scale from 0 to 3, resulting in a daily maximum score of 6. UAS7 scores are summed to yield a total score ranging from 0 to 42. A UAS7 score of 0 indicates no disease activity, while a score of 42 reflects the highest disease activity. Patients were classified into four groups: very mild (0–6); mild (7–15); moderate (16–27) and severe (28–42) [8]. Laboratory data included complete blood count with differential (eosinopenia defined as <50 cells/mL; eosinophilia > 800 cells/mL; and basopenia defined as <10 cells/mL), C-reactive protein (CRP nomal range <5 UI/mL), IgG anti-thyroid peroxidase (abnormal IgG anti-TPO >5.61 UI/mL), and total serum IgE level (normal range 40–100 UI/mL).

The Autologous Serum Skin Test (ASST) followed a protocol from an EAACI task force, requiring patients to stop antihistamines at least three days and systemic corticosteroids one

month prior [15]. Venous blood was collected, centrifuged, and the serum used for the intra-dermal test, which involved injecting 0.05 ml of the patient's serum, 0.05 ml of saline as a nega-tive control, and pricking a histamine solution as a positive control. Sites were spaced 3–5 cm apart, and results were read 30 minutes post-injection, with a positive test indicated by a serum-induced wheal $\geq 1.5$ mm more significant than the saline-induced wheal and the pres-ence of erythema. The Basophil Histamine Release Assay (BHRA) was conducted by RefLab ApS in Denmark using basophils from healthy donors [16]. The cells were treated to remove surface IgE, incubated with patient serum, and histamine release was measured. Histamine lev-els were assessed using the ortho-phthaldialdehyde method, with a response of over 16.5% considered positive.

The Chronic Urticaria Quality of Life Questionnaire (CU-Q$_2$oL) is a specialized tool designed to evaluate the impact of CSU on patients' QoL [10]. This tool has been used in stud-ies in worldwide [8, 17]. The questionnaire consists of 23 items, each rated on a 5-point Likert scale ranging from 1 (not at all) to 5 (very much). It encompasses various domains, including pruritus (itching) (2 items), swelling (2 items), life activities (6 items), sleep problems (5 items), limits (3 items), and appearance (looks) (5 items). Each domain's score is calculated by summing the responses to its constituent items, and the total CU-Q2oL score is derived by summing the scores of all domains. Higher scores indicate a more significant negative impact on quality of life.

## Statistical analysis

Descriptive statistics were used to summarize the demographic and clinical characteristics of the study population. Continuous variables were presented as means and standard deviations (SD) for normally distributed data, or medians and interquartile ranges (IQR) for non-normally distributed data. Categorical variables were presented as frequencies and percentages. Compara-tive analyses were performed using the Kruskal-Wallis test for continuous variables, depending on the data distribution. The associations between CU-Q$_2$oL scores and various clinical and demographic factors were analyzed using the multivariate Tobit regression models. Due to the small number of patients in the 'very mild' category (n = 4), we combined the 'very mild' and 'mild' categories into a single group ('$\leq$Mild [0–15]') for statistical analysis. A stepwise forward selection strategy was used to build the reduced model, with a p-value of log-likelihood test <0.2 for including variables into the models. All statistical analyses were performed using SPSS version 26.0 (IBM Corp., Armonk, NY). A p-value of less than 0.05 was considered statistically significant. The results were presented with 95% confidence intervals (CI) where applicable.

## Results

Table 1 shows that female patients constituted the majority across all severity groups (62.6%, p = 0.217). A history of allergy was present in 22.9% of patients, with no significant association with severity (p = 0.262). Acute and chronic urticaria history were reported in 12.6% and 13.7%, respectively (p = 0.755). The presence of concomitant inducible urticaria (CIndU) and angioedema were observed in 14.0% and 33.0%, respectively, with no statistical significance. Angioedema most frequently involved the lips (83.1%, p = 0.453). Itching severity showed sig-nificant differences across severity groups, with severe itching more prevalent in the severe group (38.0%, p < 0.001). Urticaria occurred daily in 80.4% of patients, predominantly in the severe group (96.1%, p < 0.001). Patients with a history of CSU treatment constituted 72.9% of the total population, with the majority responding to prior treatment (p = 0.03). Positive Autologous Serum Skin Test (ASST) and Basophil Histamine Release Assay (BHRA) results were observed in 59.5% and 10.3% of patients, respectively, without significant differences

**Table 1. Demographic, clinical and laboratory characteristics of CSU patients.**

| Characteristics | | Very Mild and Mild | Moderate | Severe | Total | p-value |
|---|---|---|---|---|---|---|
| | | N(%) | N(%) | N(%) | N(%) | |
| Sex | Female | 50 (58.8) | 38 (55.9) | 136 (66.3) | 224 (62.6) | 0.217 |
| | Male | 35 (41.2) | 30 (44.1) | 69 (33.7) | 134 (37.4) | |
| History of Allergy | | 25 (29.4) | 14 (20.6) | 43 (21.0) | 82 (22.9) | 0.262 |
| History of autoimmune disease | | 10 (11.8) | 8 (11.8) | 12 (5.9) | 30 (8.4) | 0.136 |
| History of Urticaria | None | 66 (77.6) | 52 (76.5) | 146 (71.2) | 264 (73.7) | 0.755 |
| | Acute Urticaria | 10 (11.8) | 8 (11.8) | 27 (13.2) | 45 (12.6) | |
| | Chronic Urticaria | 9 (10.6) | 8 (11.8) | 32 (15.6) | 49 (13.7) | |
| Presence of comorbid CIndU | | 14 (16.5) | 13 (19.1) | 23 (11.2) | 50 (14.0) | 0.199 |
| Presence of angioedema | | 25 (29.4) | 20 (29.4) | 73 (35.6) | 118 (33.0) | 0.467 |
| Location of angioedema (n = 118) | Eye | 17 (68.0) | 9 (45.0) | 42 (57.5) | 68 (57.6) | 0.300 |
| | Lip | 19 (76.0) | 18 (90.0) | 61 (83.6) | 98 (83.1) | 0.453 |
| | Limbs | 3 (12.0) | 3 (15.0) | 9 (12.3) | 15 (12.7) | 0.944 |
| | Throat | 1 (4.0) | 0 (0.0) | 6 (8.2) | 7 (5.9) | 0.348 |
| Severity of itching | Mild | 55 (64.7) | 17 (25.0) | 0 (0.0) | 72 (20.1) | <0.001 |
| | Moderate | 25 (29.4) | 44 (64.7) | 127 (62.0) | 196 (54.8) | |
| | Severe | 5 (5.9) | 7 (10.3) | 78 (38.0) | 90 (25.1) | |
| Number of days with urticaria per week | < 7 days | 40 (47.1) | 22 (32.4) | 8 (3.9) | 70 (19.6) | <0.001 |
| | 7 days | 45 (52.9) | 46 (67.6) | 197 (96.1) | 288 (80.4) | |
| Duration of wheal | < 1 hour | 15 (17.7) | 8 (11.8) | 19 (9.3) | 42 (11.7) | 0.282 |
| | 1–6 hours | 44 (51.8) | 39 (57.4) | 119 (58.1) | 202 (56.4) | |
| | 6–12 hours | 18 (21.2) | 18 (21.2) | 10 (14.7) | 46 (22.4) | |
| | 12- < 24 hours | 8 (9.4) | 11 (16.2) | 21 (10.2) | 40 (11.2) | |
| History of CSU treatment | No | 4 (4.7) | 7 (10.3) | 10 (4.9) | 21 (5.9) | 0.03 |
| | Unknown | 9 (10.6) | 17 (25.0) | 50 (24.4) | 76 (21.2) | |
| | Yes | 72 (84.7) | 44 (64.7) | 145 (70.7) | 261 (72.9) | |
| ASST (+) | | 45 (52.9) | 38 (55.9) | 130 (63.4) | 213 (59.5) | 0.203 |
| BHRA (+) | | 6 (7.1) | 5 (7.4) | 26 (12.7) | 37 (10.3) | 0.240 |
| Eosinopenia | | 8 (9.5) | 7 (10.5) | 32 (15.6) | 47 (13.2) | 0.290 |
| Basopenia | | 2 (2.4) | 0 (0.0) | 5 (2.4) | 7 (2.0) | 0.44 |
| | | Mean (SD) | Mean (SD) | Mean (SD) | Mean (SD) | p-value |
| Age (years) | | 36.1 (12.5) | 35.8 (14.9) | 38.9 (15.1) | 37.7 (14.5) | 0.209 |
| Duration from the first onset of urticaria (years) | | 2.1 (4.5) | 2.5 (4.9) | 3.8 (7.9) | 3.13 (6.71) | 0.159 |
| Duration of current episode (weeks) | | 34.8 (84.7) | 35.2 (56.7) | 33.5 (51.0) | 34.11 (61.49) | 0.278 |
| Eosinophil (cells/mL) | | 150.2 (108.9) | 158.9 (113.9) | 149.4 (133.9) | 151.39 (124.52) | 0.404 |
| Basophil (cells/mL) | | 45.3 (23.5) | 39.8 (21.1) | 46.7 (43.1) | 45.05 (35.84) | 0.272 |
| CRP (UI/mL) | | 2.1 (2.9) | 2.3 (3.0) | 2.7 (5.9) | 2.46 (4.86) | 0.724 |
| Total serum IgE (UI/mL) | | 304.0 (429.1) | 289.4 (272.3) | 333.8 (414.4) | 318.32 (394.64) | 0.659 |
| IgG anti-TPO (UI/mL) | | 20.0 (114.1) | 38.4 (173.8) | 24.3 (127.5) | 25.96 (134.40) | 0.888 |

Abbrev: BHRA: Basophil Histamine Release Assay; UAS7: Urticaria Activity Score; ASST: Autologous Serum Skin Test; CIndU: Chronic inducible urticaria; CSU: Chronic Spontaneous Urticaria

across groups. Eosinopenia and basopenia were present in 13.2% and 2.0% of patients, respectively, with no significant associations. Mean age, duration of urticaria episodes, and other laboratory parameters such as CRP and total serum IgE levels did not differ significantly between severity groups.

**Table 2. Disease activity and quality of life among CSU patients.**

| Characteristics | Frequency (n = 358) | Percentage (%) |
|---|---|---|
| **UAS7 categories** | | |
| Very mild (0–6) | 4 | 0.8 |
| Mild (7–15) | 81 | 22.6 |
| Moderate (16–27) | 68 | 19.0 |
| Severe (28–42) | 205 | 57.3 |
| | **Mean (SD)** | **Median (IQR)** |
| UAS7 | 25.54 (9.62) | 28 (17–32) |
| CU-Q$_2$oL Total | 48.67 (16.90) | 46 (35–59) |
| CU-Q$_2$oL Pruritus | 5.42 (2.02) | 5 (4–7) |
| CU-Q$_2$oL Swelling | 2.86 (1.54) | 2 (2–3) |
| CU-Q$_2$oL Life activities | 13.89 (6.00) | 14 (9–18) |
| CU-Q$_2$oL Sleep problems | 11.12 (4.96) | 10 (7–15) |
| CU-Q$_2$oL Limits | 6.52 (2.66) | 6 (5–8) |
| CU-Q$_2$oL Looks | 8.85 (4.09) | 8 (5–11) |

Abbrev: UAS7: Urticaria Activity Score in 7 days

Table 2 shows that the mean UAS7 score was 25.54 (SD = 9.62), with a median of 28 (IQR = 17–32). The majority of patients (57.3%) fell into the severe UAS7 category (28–42), with 19.0% in the moderate category (16–27), 22.6% in the mild category (7–15), and only 0.8% in the very mild category (0–6).

The CU-Q$_2$oL total score had a mean of 48.67 (SD = 16.90) and a median of 46 (IQR = 35–59). The mean scores for individual CU-Q$_2$oL subscales were as follows: pruritus (5.42±2.02), swelling (2.86±1.54), life activities (13.89±6.00), sleep problems (11.12±4.96), limits (6.52 ±2.66), and looks (8.85±4.09).

Fig 1 illustrates that the "Very mild (0–6)" group had a mean score of 4.67 (SD = 1.15). The "Mild (7–15)" group had a mean score of 12.81 (SD = 2.11). The "Moderate (16–27)" group had a mean score of 20.65 (SD = 2.20). The "Severe (28–42)" group had a mean score of 32.49 (SD = 5.57).

The CU-Q$_2$oL scores across different UAS7 categories are presented in Fig 2A. The "Very mild and Mild (0–15)", "Moderate (16–27)" and "Severe (28–42)" groups had a mean score of 41.7 (SD = 10.1), 48.5 (SD = 12.2), 51.4 (SD = 15.3), respectively, with the differences being statistically significant (p<0.01). Fig 2B breaks down the CU-Q2oL domains. The CU-Q$_2$oL scores across different domains varied significantly among the ≤Mild, Moderate, and Severe UAS7 categories. In the Pruritus domain, mean scores were 4.6 (SD = 1.3), 5.5 (SD = 1.5), and 5.7 (SD = 1.8), respectively. Similarly, the Swelling domain showed mean scores of 2.5 (SD = 1.2), 2.7 (SD = 1.3), and 3.1 (SD = 1.5). For Life activities, the scores were 12.4 (SD = 4.2), 13.2 (SD = 4.5), and 14.7 (SD = 5.0), while the Sleep problems domain had mean scores of 9.2 (SD = 3.1), 11.4 (SD = 3.6), and 11.8 (SD = 3.9). The Limits domain recorded mean scores of 5.6 (SD = 2.0), 6.8 (SD = 2.5), and 6.9 (SD = 2.7), and in the Looks domain, the scores were 7.4 (SD = 2.6), 8.9 (SD = 3.1), and 9.4 (SD = 3.3), respectively. All differences of QoL domain scores across UAS7 categories were significant (p<0.01).

Table 3 presents the factors associated with CU-Q$_2$oL total and domain scores among CSU patients. Age was negatively correlated with CU-Q$_2$oL Looks (Coef = -0.06, 95%CI: -0.10, -0.02). Positive ASST status was negatively related to CU-Q$_2$oL Total (Coef = -4.71, 95%CI: -8.29, -1.13), CU-Q$_2$oL Swelling (Coef = -2.09, 95%CI: -3.49, -0.69) and CU-Q$_2$oL Limits

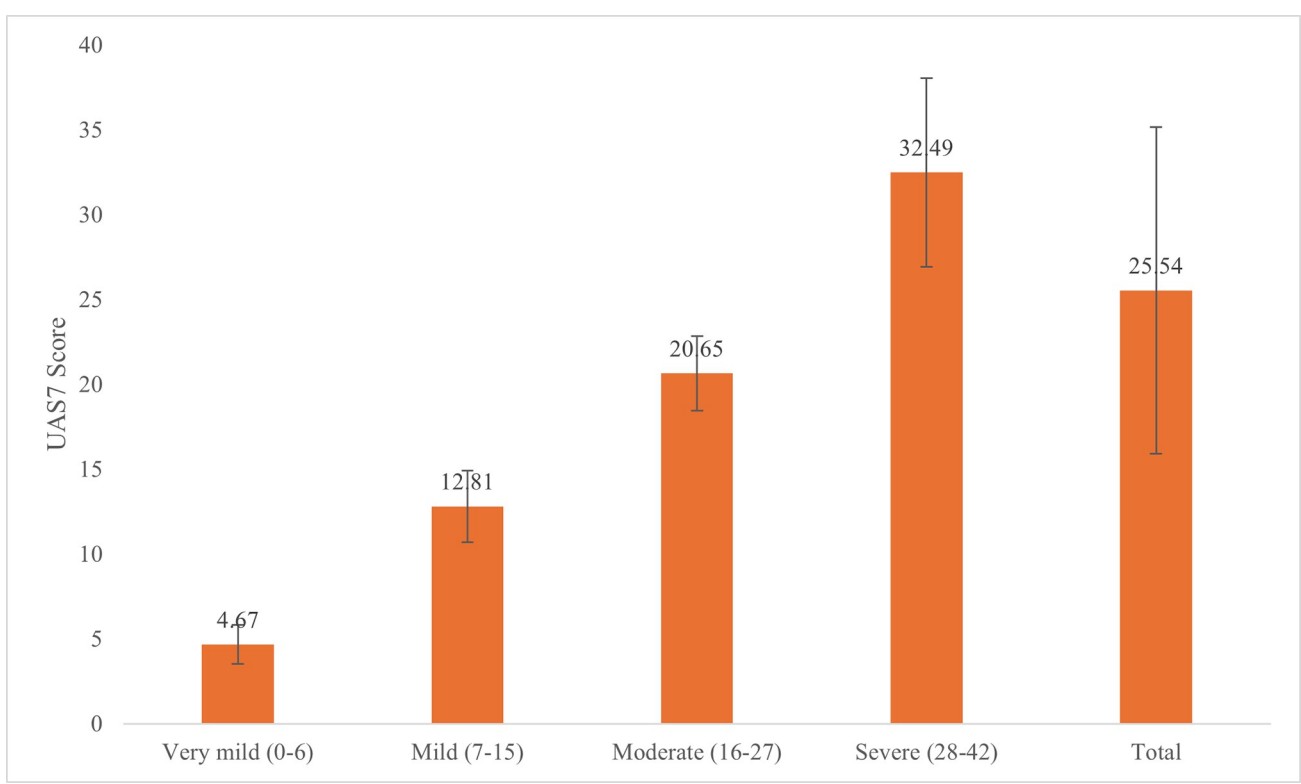

**Fig 1. Mean urticaria activity score over 7 days across UAS7 categories.**

(Coef = -1.18, 95%CI: -1.83, -0.53). Positive BHRA status was negatively related to CU-$Q_2$oL Sleep problems (Coef = -2.53, 95%CI: -4.57, -0.50).

Duration from the first onset of urticaria was positively correlated with CU-$Q_2$oL Pruritus (Coef = 0.03, 95%CI: 0.00, 0.07). A history of allergy was positively connected with CU-$Q_2$oL Total (Coef = 4.67, 95%CI: 0.62, 8.72), CU-$Q_2$oL Sleep problems (Coef = 1.46, 95%CI: 0.05, 2.86), and CU-$Q_2$oL Looks (Coef = 1.68, 95%CI: 0.38, 2.99). Severe itching was positively linked with CU-$Q_2$oL Sleep problems (Coef = 2.86, 95%CI: 0.56, 5.17).

Higher levels of UAS7 categories were positively associated with CU-$Q_2$oL total and all domain scores. Angioedema in the eyes was positively related to CU-$Q_2$oL Swelling (Coef = 1.96, 95%CI: 0.94;2.98). Meanwhile, the presence of angioedema in the lip was positively linked with CU-$Q_2$oL Total (Coef = 4.17, 95%CI: 0.35, 7.99) and CU-$Q_2$oL Swelling (Coef = 2.12, 95%CI: 1.17, 3.07).

Basopenia was positively related to CU-$Q_2$oL Swelling (Coef = 2.77, 95%CI: 0.29, 5.26), while eosinopenia was negatively correlated with CU-$Q_2$oL Life activities (Coef = -2.07, 95% CI: -4.13, -0.01).

## Discussion

This study evaluates the QoL in patients with CSU in Vietnam. The findings underscore the significant burden of CSU on patients' daily lives, highlighting the impact on various QoL domains such as pruritus, swelling, life activities, sleep problems, limits, and appearance. Different clinical and laboratory characteristics of patients were associated with QoL of CSU patients, suggesting further implications for improving CSU management in the clinical setting.

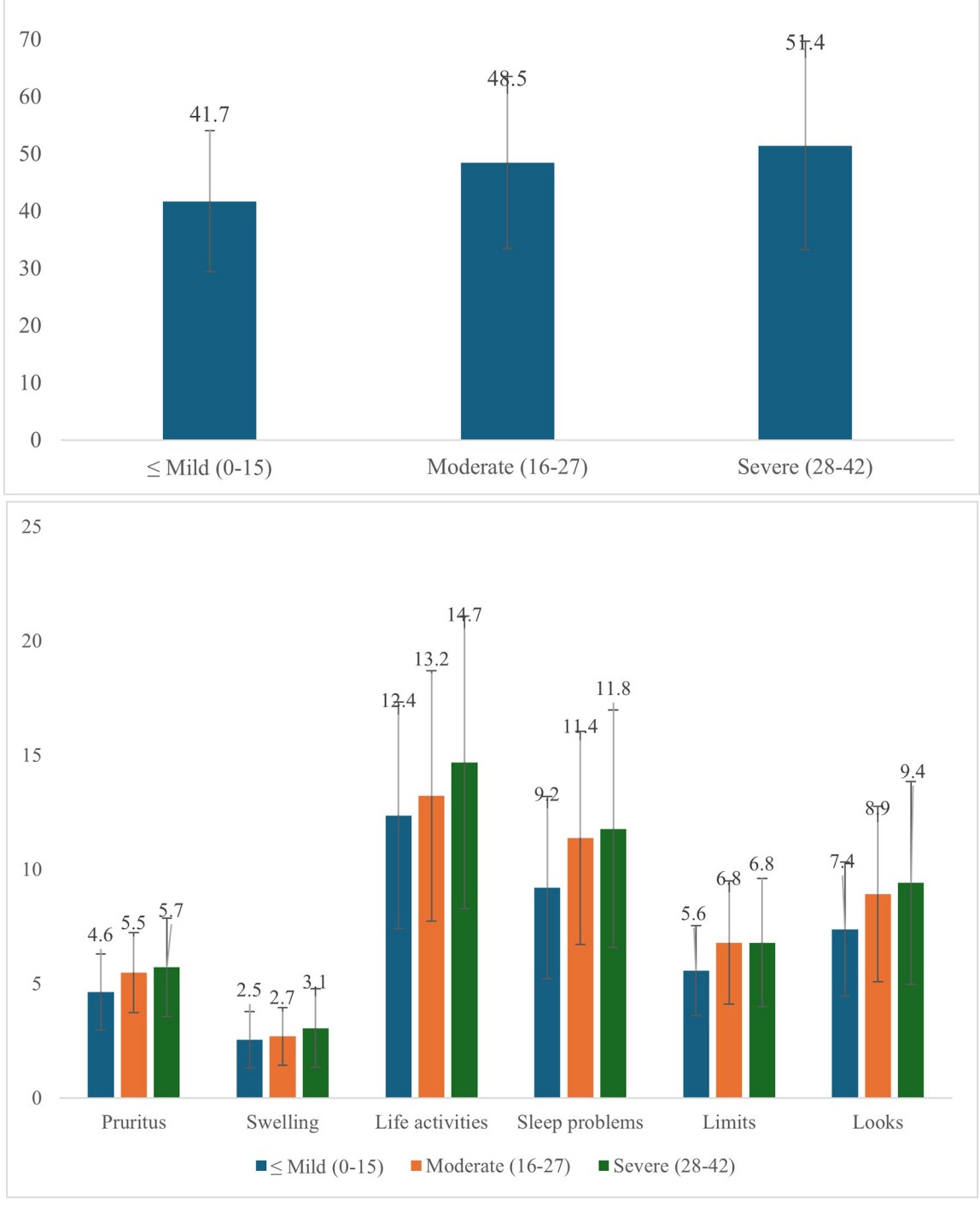

**Fig 2.** CU-Q$_2$oL a) total and b) domains across UAS7 categories.

The impact of CSU on the QoL of patients is profound and multifaceted. One of the most significant impacts of CSU is the severity of itching, or pruritus, which significantly affects patients' QoL. Itching causes not only physical discomfort but also psychological distress, as

**Table 3. Factors associated with CU-Q$_2$oL total and domains among CSU patients.**

| Characteristics | CU-Q$_2$oL Total | | CU-Q$_2$oL Pruritus | | CU-Q$_2$oL Swelling | | CU-Q$_2$oL Life activities | | CU-Q$_2$oL Sleep problems | | CU-Q$_2$oL Limits | | CU-Q$_2$oL Looks | |
|---|---|---|---|---|---|---|---|---|---|---|---|---|---|---|
| | Coef. | 95%CI | Coef. | 95%CI | Coef. | 95%CI | Coef. | 95%CI | Coef. | 95%CI | Coef. | 95%CI | Coef. | 95%CI |
| **Age (per year)** | | | -0.01 | -0.03; 0.00 | | | | | | | -0.02 | -0.04; 0.00 | -0.06* | -0.10; -0.02 |
| **Gender** | | | | | | | | | | | | | | |
| Female | | | | | ref | | | | ref | | ref | | | |
| Male | | | | | -0.60 | -1.41; 0.22 | | | -0.84 | -2.09; 0.40 | 0.60 | -0.03; 1.24 | | |
| **Duration from the first onset of urticaria (per year)** | | | 0.03* | 0.00; 0.07 | | | | | | | 0.04 | -0.01; 0.08 | | |
| **Duration of the current episode (per week)** | | | -0.00 | -0.01; 0.00 | | | -0.01 | -0.02; 0.00 | | | | | | |
| **History of allergy** | | | | | | | | | | | | | | |
| No | ref | | | | | | ref | | | | | | ref | |
| Yes | 4.67* | 0.62; 8.72 | | | | | 1.27 | -0.36; 2.90 | 1.46* | 0.05; 2.86 | | | 1.68* | 0.38; 2.99 |
| **History of CSU treatment** | | | | | | | | | | | | | | |
| No | | | | | | | | | | | ref | | ref | |
| Yes | | | | | | | | | | | 0.45 | -0.08; 0.98 | 0.92 | -0.07; 1.90 |
| **Itching severity** | | | | | | | | | | | | | | |
| Mild | ref | | | | | | | | ref | | ref | | ref | |
| Moderate | 1.11 | -4.56; 6.78 | | | | | | | 1.57 | -0.41; 3.55 | 0.22 | -0.78; 1.23 | -0.35 | -2.21; 1.51 |
| Severe | 4.83 | -1.82; 11.48 | | | | | | | 2.86* | 0.56; 5.17 | 1.10 | -0.09; 2.28 | 1.35 | -0.80; 3.50 |
| **UAS7 categories** | | | | | | | | | | | | | | |
| ≤ Mild | ref | | ref | | ref | | ref | | ref | | ref | | ref | |
| Moderate | 7.32* | 1.66; 12.98 | 0.93* | 0.25; 1.61 | 0.76 | -0.52; 2.03 | 1.12 | -1.00; 3.23 | 2.36* | 0.39; 4.32 | 1.38* | 0.37; 2.39 | 2.84* | 0.98; 4.71 |
| Sever | 9.05* | 3.46; 14.65 | 1.13* | 0.58; 1.67 | 1.37* | 0.35; 2.38 | 2.93* | 1.24; 4.61 | 1.89 | -0.05; 3.84 | 1.05* | 0.05; 2.06 | 3.07* | 1.23; 4.91 |
| **Presence of comorbid CIndU** | | | | | | | | | | | | | | |
| No | | | | | | | | | | | ref | | | |
| Yes | | | | | | | | | | | -0.62 | -1.54; 0.29 | | |
| **Presence of angioedema in the eye** | | | | | | | | | | | | | | |
| No | | | | | ref | | | | | | | | | |
| Yes | | | | | 1.96* | 0.94; 2.98 | | | | | | | | |
| **Presence of angioedema in lip** | | | | | | | | | | | | | | |
| No | ref | | ref | | ref | | ref | | | | ref | | ref | |
| Yes | 4.17* | 0.35; 7.99 | 0.36 | -0.14; 0.86 | 2.12* | 1.17; 3.07 | 1.18 | -0.38; 2.74 | | | 0.53 | -0.18; 1.23 | 0.80 | -0.42; 2.03 |
| **Presence of angioedema in limbs** | | | | | | | | | | | | | | |
| No | | | | | | | | | | | ref | | | |

(*Continued*)

**Table 3.** (Continued)

| Characteristics | CU-Q$_2$oL Total | | CU-Q$_2$oL Pruritus | | CU-Q$_2$oL Swelling | | CU-Q$_2$oL Life activities | | CU-Q$_2$oL Sleep problems | | CU-Q$_2$oL Limits | | CU-Q$_2$oL Looks | |
|---|---|---|---|---|---|---|---|---|---|---|---|---|---|---|
| | Coef. | 95%CI | Coef. | 95%CI | Coef. | 95%CI | Coef. | 95%CI | Coef. | 95%CI | Coef. | 95%CI | Coef. | 95%CI |
| Yes | | | | | | | | | | | -1.51 | -3.14; 0.13 | | |
| **ASST** | | | | | | | | | | | | | | |
| Negative | ref | | | | | | ref | | ref | | ref | | ref | |
| Positive | -4.71* | -8.29; -1.13 | | | | | -2.09* | -3.49; -0.69 | -1.11 | -2.37; 0.15 | -1.18* | -1.83; -0.53 | -0.86 | -1.98; 0.26 |
| **BHRA** | | | | | | | | | | | | | | |
| Negative | ref | | | | | | | | ref | | | | | |
| Positive | -4.39 | -10.18; 1.40 | | | | | | | -2.53* | -4.57; -0.50 | | | | |
| **Basopenia** | | | | | | | | | | | | | | |
| No | ref | | ref | | ref | | | | ref | | | | | |
| Yes | 9.36 | -2.85; 21.57 | 1.51 | -0.09; 3.12 | 2.77* | 0.29; 5.26 | | | 3.97 | -0.24; 8.18 | | | | |
| **Eosinopenia** | | | | | | | | | | | | | | |
| No | | | | | | | ref | | | | | | | |
| Yes | | | | | | | -2.07* | -4.13; -0.01 | | | | | | |

*p<0.05; Abbrev: BHRA: Basophil Histamine Release Assay; UAS7: Urticaria Activity Score; ASST: Autologous Serum Skin Test; CIndU: Chronic inducible urticaria; CSU: Chronic Spontaneous Urticaria

the itching associated with CSU can result in fatigue and soreness, leading to severe sleep disturbances [9, 18, 19]. Our study found that severe itching was associated with higher scores in the domain of sleep problems, reflecting the widespread disruption of rest and recuperation. Sleep disturbances are a significant concern for CSU patients, as itching and discomfort often disrupt sleep patterns, leading to poor sleep quality and chronic fatigue [9]. This can have a cascading effect on overall health, cognitive function, and emotional well-being.

Angioedema, another hallmark of CSU, exacerbates the disease burden, mainly when swelling occurs in visible areas like the face, such as the eyes and lips [13]. This swelling can lead to social embarrassment and significant emotional distress, contributing to high scores in the CU-Q$_2$oL swelling domain [13]. Our finding was consistent with prior research, which showed that angioedema had negative impacts on patients' QoL [20, 21]. The visibility of these symptoms often results in patients avoiding social interactions and activities, leading to social isolation and diminished life satisfaction. The social and emotional repercussions of angioedema highlight the need for targeted interventions to manage these symptoms effectively [9].

CSU also significantly hampers daily activities and routines, as reflected in the high scores of the life activities domain of the CU-Q$_2$oL. The condition interferes with work, physical activities, and social engagements, forcing patients to adjust their daily routines to manage symptoms and avoid triggers that could worsen their condition [9]. These limitations on lifestyle choices and independence contribute to frustration and helplessness, negatively impacting patients' QoL.

The Limits domain, encompassing restrictions on eating or sports activities, also scored high, indicating that CSU frequently limits patients' ability to participate in various activities. These limitations can lead to frustration, helplessness, and reduced self-esteem, affecting work productivity and personal relationships, further diminishing QoL. Patients with chronic

spontaneous urticaria often experience a higher prevalence of depression, anxiety, and conflicts within their social environment [22]. The effects of chronic spontaneous urticaria extend beyond physical symptoms, affecting various aspects of life, including work performance, leisure activities, and family relationships [23]. This chronic condition can be unpredictable and disabling, impairing mobility, work management, and overall well-being [24].

The Looks domain of the CU-$Q_2$oL is particularly relevant in CSU, as the visible symptoms of swelling and angioedema impact self-image and confidence. Age negatively correlated with the "Looks" domain of CU-$Q_2$oL, suggesting that younger patients may be more affected by the appearance-related aspects of CSU. This finding is similar to the prior studies, which indicated that young people perceived more negative impacts of the illness [11, 25, 26]. Younger patients, in particular, maybe more affected by these visible signs of the disease, influencing social interactions and psychological health.

Our study identified several factors significantly associated with QoL in CSU patients. Positive ASST status and BHRA were negatively associated with CU-$Q_2$oL scores and specific domains such as sleep, swelling, and limits, indicating that autoimmunity may play a role in worsening QoL. Chronic spontaneous urticaria with autoimmune characteristics often has symptoms that worsen at night, which partly explains its impact on sleep [27]. Studies have shown that patients with CSU, especially those with positive ASST results, tend to experience more severe clinical manifestations and have a lower quality of life compared to those with negative ASST results [23]. The clinical manifestations of CSU patients with positive and negative ASST results may appear similar, but the impact on QoL can be notably different [28].

Basopenia was positively related to CU-$Q_2$oL swelling. Basopenia, characterised by decreased peripheral basophil counts, has been observed in CSU patients and has implications for their QoL. Studies have shown that basopenia is more frequently associated with CSU and is linked to disease activity [29, 30]. The phenomenon of basopenia during the active phase of urticaria has been confirmed, and basophil numbers tend to increase following symptom improvement in patients treated with medications like omalizumab and antihistamines [31]. Basopenia is more prevalent in CSU patients and is associated with higher baseline urticaria activity scores [30]. Additionally, basopenia has been correlated with disease severity, as reduced IgE-mediated basophil histamine release and basopenia are observed in active CSU [32], which might be related to the reduction of QoL.

The findings of this study have several implications for clinical practice and healthcare policy. Firstly, the substantial impact of CSU on QoL highlights the necessity for comprehensive management strategies that address the condition's physical and psychological aspects. Healthcare providers should consider incorporating regular QoL assessments into routine clinical practice to understand CSU patients better and address their needs. Additionally, the study underscores the importance of personalised treatment approaches that consider Vietnam's unique socio-cultural, economic, and healthcare dynamics. Developing culturally appropriate management strategies and policies is crucial for improving patient care and outcomes.

This study has several limitations. The cross-sectional design limits the ability to establish causality between CSU and QoL. Longitudinal studies are needed to understand better the temporal relationship between QoL and its associated factors. The study was conducted at a single center, which may limit the generalizability of the findings to other settings in Vietnam. Future studies should consider including multiple centers to enhance the sample's representativeness. Additionally, the reliance on patient self-reports for specific clinical characteristics may introduce recall bias, although standardised data collection procedures were employed to mitigate this issue. While the CU-Q2oL has been proven effective in various global studies, its adequacy for the Vietnamese setting has not been specifically validated. Given the unique socio-cultural and healthcare dynamics in Vietnam, minor modifications or adaptations might

be necessary to ensure the tool's relevance and cultural appropriateness. Further research would be beneficial to assess whether the existing version fully captures the QoL dimensions important to this population or if localized adaptations are required.

## Conclusion

In conclusion, this study provides valuable insights into the QoL of CSU patients in Vietnam, highlighting the significant burden of the condition on various aspects of daily life. The findings underscore the necessity of comprehensive and culturally appropriate management strategies to address CSU's physical and psychological impacts. Identifying factors associated with poorer QoL can inform targeted interventions to improve patient outcomes. Future research should focus on longitudinal studies and multi-centre collaborations to further elucidate the long-term impact of CSU and enhance the generalizability of findings.

## Author Contributions

**Conceptualization:** Nguyen Thi Tra My, Vu Nguyet Minh, Doanh Le Huu.

**Investigation:** Nguyen Thi Tra My, Vu Nguyet Minh, Nguyen Thi Ha Vinh.

**Methodology:** Nguyen Thi Tra My, Le Huyen My, Vu Nguyet Minh, Doanh Le Huu.

**Project administration:** Nguyen Thi Tra My, Le Huyen My, Nguyen Thi Ha Vinh, Mai Ba Hoang Anh.

**Software:** Le Huyen My, Mai Ba Hoang Anh.

**Supervision:** Le Huyen My, Vu Nguyet Minh, Doanh Le Huu.

**Validation:** Nguyen Thi Ha Vinh, Mai Ba Hoang Anh, Doanh Le Huu.

**Writing – original draft:** Nguyen Thi Tra My, Le Huyen My, Vu Nguyet Minh, Nguyen Thi Ha Vinh, Mai Ba Hoang Anh, Doanh Le Huu.

**Writing – review & editing:** Nguyen Thi Tra My, Le Huyen My, Vu Nguyet Minh, Nguyen Thi Ha Vinh, Mai Ba Hoang Anh, Doanh Le Huu.

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
