## [Decision Letter · Decision Letter 0]

25 Sep 2024

PONE-D-24-32836Factors Associated with Quality of Life of Chronic Spontaneous Urticaria Patients in a Vietnamese PopulationPLOS ONE

Dear Dr. Le,

Thank you for submitting your manuscript to PLOS ONE. After careful consideration, we feel that it has merit but does not fully meet PLOS ONE’s publication criteria as it currently stands. Therefore, we invite you to submit a revised version of the manuscript that addresses the points raised during the review process.

**ACADEMIC EDITOR: **The editorial board and external reviewers have thoroughly evaluated your manuscript. It may be reconsidered if appropriate revisions are made. Specifically, the study methods and data analysis need to be presented more clearly.

We look forward to receiving your revised manuscript.

Kind regards,

Bharat Bhushan Sharma, M.D.

Academic Editor

PLOS ONE

Additional Editor Comments:

The editorial board and external reviewers have thoroughly evaluated your manuscript. It may be reconsidered if appropriate revisions are made. Specifically, the study methods and data analysis need to be presented more clearly.

Reviewers' comments:

Reviewer's Responses to Questions

**Comments to the Author**

1. Is the manuscript technically sound, and do the data support the conclusions?

Reviewer #1: Yes

Reviewer #2: Yes

2. Has the statistical analysis been performed appropriately and rigorously? 

Reviewer #1: Yes

Reviewer #2: Yes

3. Have the authors made all data underlying the findings in their manuscript fully available?

Reviewer #1: Yes

Reviewer #2: Yes

4. Is the manuscript presented in an intelligible fashion and written in standard English?

Reviewer #1: Yes

Reviewer #2: Yes

5. Review Comments to the Author

Reviewer #1: The manuscript entitled 'Factors associated with quality of life of chronic spontaneous urticaria patients in a Vietnamese population' by My et al addressed the question about affecting factors on QOL of Vietnamese people. To do so, they employed cross-sectional study and conducted at the Vietnam National Dermatology and Venereology Hospital by recruiting 358 patients with chronic spontaneous urticaria aged older than 16 years old. Consequently, they found a panel of factors negatively affecting the patients' QSL (as shown in Table 2 and 3). Although this kind of analysis is not unique, we clinician should know the ethnic difference regarding each disease. In this context, it is worth reading in order to know Vietnamese mentality on this chronic pruritic allergic disorder.

Reviewer #2: The authors describe data on chronic spontaneous urticaria in a Vietnamese population. They assess QoL in these patients and correlate QoL responses with different conditions.

Some issues to address:

- Do the authors consider the QoL questionnaire adequate for the Vietnamese setting? Would the authors propose any modifications or an adapted version? They should discuss whether or not the QoL questionnaire is compatible with the Vietnemese population or which modifications could be suggested

- The diagnostic criteria of chronic spontaneous urticaria could be clearer. Did the authors investigate aggravating factors such as NSAIDs? Newly administered drugs or infections were actively excluded?

- The authors mention basopenia and eosinopenia, but there is no mention of eosinophilia in the studied population. Were there eosinophilic patients in the cohort? Were causes of eosinophilia such as atopic or parasitic disease excluded?

- Do the authors have data on systemic symptoms such as fever and arthralgia? These symptoms can be correlated with urticaria and impact patient’s QoL.

6. PLOS authors have the option to publish the peer review history of their article (what does this mean?). If published, this will include your full peer review and any attached files.

Reviewer #1: No

Reviewer #2: No

---

## [Author Response · Author response to Decision Letter 0]

5 Oct 2024

Reviewer #1

Comment: The manuscript addresses the factors affecting QoL in Vietnamese patients with chronic spontaneous urticaria, which is worth reading for understanding ethnic differences. 

Response: We appreciate the positive feedback and agree that understanding ethnic differences in chronic spontaneous urticaria is essential for improving patient care. No further modifications were made based on this comment.

Reviewer #2

1) Comment: Do the authors consider the QoL questionnaire adequate for the Vietnamese setting? Would any modifications or adaptations be needed? 

Response: We have revised the manuscript to include a discussion about the appropriateness of the QoL questionnaire for the Vietnamese population. The questionnaire used has been validated in several settings, including Vietnam, but we have highlighted that future studies could explore cultural adaptations to improve the sensitivity of the tool for the Vietnamese population. 

Position: Limitations paragraph in Discussion section

2) Comment: The diagnostic criteria of chronic spontaneous urticaria could be clearer. Did the authors investigate aggravating factors like NSAIDs, newly administered drugs, or infections? Were these actively excluded? 

Response: We have clarified the diagnostic criteria for chronic spontaneous urticaria in the Methods section. Inclusion criteria required a diagnosis of CSU based on EAACI guidelines, characterized by the spontaneous appearance of wheals (hives) or angioedema for six weeks or more without any identifiable trigger factors. Wheals were defined as red or pink raised areas lasting less than 24 hours, while angioedema involved swelling of subcutaneous and submucosal tissues lasting up to 72 hours. Symptoms had to occur daily or almost daily for more than six weeks. Patients who were using NSAIDs were required to discontinue them at least one week prior to performing the Autologous Serum Skin Test (ASST), along with antihistamines. Additionally, patients exhibiting abnormal clinical, or laboratory signs were further screened to identify underlying causes. For instance, patients with fever or arthralgia were screened for infections, while those with eosinophilia underwent screening for parasitic infections. Any cases where a specific cause for the urticaria was identified were excluded from the study. This process ensured that the study focused only on patients with true CSU, free from external aggravating factors

Position: Method

3) Comment: The authors mention basopenia and eosinopenia but do not mention eosinophilia. Were eosinophilic patients present? Were causes of eosinophilia, such as atopic or parasitic disease, excluded? 

Response: In our study, we detected only one patient with eosinophilia. We have provided data in Table 1. Patients with eosinophilia underwent screening for parasitic infections. Any cases where a specific cause for the urticaria was identified were excluded from the study. This process ensured that the study focused only on patients with true CSU, free from external aggravating factors. 

Position: Method

4) Comment: Do the authors have data on systemic symptoms such as fever and arthralgia, which can correlate with urticaria and impact QoL? 

Response: Systemic symptoms such as fever and arthralgia are typically associated with acute urticaria caused by infections, rather than being symptoms of chronic spontaneous urticaria (CSU). If a patient with CSU presents with a fever, tests should be conducted to identify the underlying cause of the fever, as it is likely due to a concurrent condition and not a symptom of CSU. For example, there may be cases of dengue fever occurring during the course of CSU treatment, but not simultaneously with the onset of CSU. In our study, we did not record any cases of fever or arthralgia.

---

## [Decision Letter · Decision Letter 1]

2 Dec 2024

PONE-D-24-32836R1Factors Associated with Quality of Life of Chronic Spontaneous Urticaria Patients in a Vietnamese PopulationPLOS ONE

Dear Dr. Le,

Thank you for submitting your manuscript to PLOS ONE. After careful consideration, we feel that it has merit but does not fully meet PLOS ONE’s publication criteria as it currently stands. Therefore, we invite you to submit a revised version of the manuscript that addresses the points raised during the review process.

**ACADEMIC EDITOR: **After the initial review and the author's subsequent revisions, the journal's requirements necessitated an additional review, which has now been completed. The comments are being forwarded for further amendments. ==============================

We look forward to receiving your revised manuscript.

Kind regards,

Bharat Bhushan Sharma, M.D.

Academic Editor

PLOS ONE

Journal Requirements:

Additional Editor Comments:

After the initial review and the author's subsequent revisions, the journal's requirements necessitated an additional review, which has now been completed. The comments are being forwarded for further amendments.

Reviewers' comments:

Reviewer's Responses to Questions

**Comments to the Author**

1. If the authors have adequately addressed your comments raised in a previous round of review and you feel that this manuscript is now acceptable for publication, you may indicate that here to bypass the “Comments to the Author” section, enter your conflict of interest statement in the “Confidential to Editor” section, and submit your "Accept" recommendation.

Reviewer #1: All comments have been addressed

Reviewer #3: (No Response)

Reviewer #4: (No Response)

2. Is the manuscript technically sound, and do the data support the conclusions?

Reviewer #1: Yes

Reviewer #3: Yes

Reviewer #4: Partly

3. Has the statistical analysis been performed appropriately and rigorously? 

Reviewer #1: Yes

Reviewer #3: Yes

Reviewer #4: Yes

4. Have the authors made all data underlying the findings in their manuscript fully available?

Reviewer #1: Yes

Reviewer #3: No

Reviewer #4: Yes

5. Is the manuscript presented in an intelligible fashion and written in standard English?

Reviewer #1: Yes

Reviewer #3: Yes

Reviewer #4: Yes

6. Review Comments to the Author

Reviewer #1: As I mentioned to the authors, this manuscript is worth being accepted in PLoS One, based on the face that it's important to know ethnic feelings to the diseases such as urticaria even when similar analyses were already published before.

Reviewer #3: The study is very interesting and will contribute to the understanding of the factors affecting QoL in chronic spontaneous urticaria. It may help in enhancing management strategies involving CSU. The article is carefully written, methodology is robust and the research questions have been adequately addressed. I have provided my comments below.

Abstract:

1. To improve readability, please restructure this part: ‘…in the eyes and lips worsened swelling and overall QoL’.

2. “Longer disease duration increased pruritus scores, while a history of allergy worsened total QoL, sleep, and looks”. Please consider using the words like ‘associated’ or ‘related’ in this context.

Introduction:

3. Please provide a recent reference supporting the estimated point prevalence of CSU. (Line 4; …estimated point prevalence of around 0.5% to 1%.)

4. Please maintain consistency in using abbreviated form. For example, CSU or CU. (Line 18; … for CU patients has been..)

Materials and Method:

5. Please keep either the words ‘CSU patients’ or ‘with CSU’ in the sentence ‘A cross-sectional, single-center study involving CSU patients aged 16 years or older with CSU was conducted at the Urticaria Clinic at the Vietnam National Dermatology and Venereology Hospital from 1 June 2023 to 31 March 2024.’

6. How the severity of itching (mild/ moderate/ severe) was assessed? Did you use any scale? You have also evaluated pruritus (2 items) as a part of CU-Q2oL tool. Was it done differently?

Results:

7. Though mentioned 358, the sum of n in table 2 in actually 357. If there was a missing value, please mention.

8. ‘The “≤Mild (0-15)”, “Moderate (16-27)” and “Severe (28-42)” groups had…’. It looks like that, you actually combined ‘very mild’ and ‘mild’ together and considered those as ‘mild’. This may be acceptable, because ‘very mild’ group had only 3 participants. In addition, some in ‘very mild’ UAS7 category might also be assessed as ‘mild’ category. Please add a sentence in the method section addressing this issue about combining those two.

Conclusion:

9. You may consider replacing the word ‘importance’ with the word ‘necessity’ in the sentence, “The findings underscore the importance of comprehensive and culturally appropriate management strategies to address CSU's physical and psychological impacts.”

Reviewer #4: In the manuscript, the authors explored the quality of life (QoL) among patients with chronic spontaneous urticaria (CSU) in Vietnam and identified factors associated with QoL. This cross-sectional study recruited 358 patients aged 16 years or older at the Vietnam National Dermatology and Venereology Hospital between June 2023 and March 2024. Using the Chronic Urticaria Quality of Life Questionnaire (CU-Q2oL) and the Weekly Urticaria Activity Score (UAS7), the study assessed QoL and disease severity. The findings revealed significant impairment in QoL across physical, emotional, and social domains, with factors such as higher UAS7 scores, angioedema in the eyes and lips, longer disease duration, and a history of allergy being associated with poorer QoL outcomes. These findings emphasize the importance of integrating regular QoL assessments into CSU management to support patient-centered care strategies.

The authors have appropriately responded to the reviewers' comments, enhancing the clarity and rigor of their findings. However, this study is a retrospective, single-center observational analysis, necessitating careful attention to comparative analyses of the data. In particular, comparisons among UAS7 mild, UAS7 moderate, and UAS7 severe groups require meticulous consideration. While Table 1 provides baseline characteristics, it would be beneficial to stratify these into the three UAS7 severity groups and assess for any statistically significant differences. Even if no statistically significant differences are identified, any apparent variation between groups warrants additional discussion regarding the potential for confounding bias. This further analysis would strengthen the study's validity and ensure comprehensive interpretation of the findings.

7. PLOS authors have the option to publish the peer review history of their article (what does this mean?). If published, this will include your full peer review and any attached files.

Reviewer #1: No

Reviewer #3: **Yes: **Ruhul Amin

Reviewer #4: No

---

## [Author Response · Author response to Decision Letter 1]

3 Dec 2024

Dear Editor,

We appreciate the constructive feedback provided by the reviewers on our manuscript titled "Factors Associated with Quality of Life of Chronic Spontaneous Urticaria Patients in a Vietnamese Population." Below, we provide a detailed response to each comment.

Reviewer #3

Abstract:

Comment 1:

To improve readability, please restructure this part: “…in the eyes and lips worsened swelling and overall QoL.”

Response:

We have revised the sentence for clarity:

Original: “…in the eyes and lips worsened swelling and overall QoL.”

Revised: “Higher UAS7 scores were associated with lower QoL, and angioedema in the eyes and lips were associated with increased swelling and poorer overall QoL.”

Comment 2:

“Longer disease duration increased pruritus scores, while a history of allergy worsened total QoL, sleep, and looks.” Please consider using the words like ‘associated’ or ‘related’ in this context.

Response:

The sentence has been updated for precision:

Original: “Longer disease duration increased pruritus scores, while a history of allergy worsened total QoL, sleep, and looks.”

Revised: “Longer disease duration was associated with higher pruritus scores, while a history of allergy was related to poorer total QoL, sleep, and looks.”

Introduction:

Comment 3:

Please provide a recent reference supporting the estimated point prevalence of CSU (Line 4; “…estimated point prevalence of around 0.5% to 1%.”).

Response:

We have updated the recent references

Comment 4:

Please maintain consistency in using abbreviated form. For example, CSU or CU (Line 18; “…for CU patients has been…”).

Response:

We have ensured consistent use of “CSU” throughout the manuscript.

Materials and Method:

Comment 5:

Please keep either the words ‘CSU patients’ or ‘with CSU’ in the sentence “A cross-sectional, single-center study involving CSU patients aged 16 years or older with CSU was conducted at the Urticaria Clinic at the Vietnam National Dermatology and Venereology Hospital from 1 June 2023 to 31 March 2024.”

Response:

We have restructured the sentence:

Original: “A cross-sectional, single-center study involving CSU patients aged 16 years or older with CSU was conducted...”

Revised: “A cross-sectional, single-center study involving CSU patients aged 16 years or older was conducted...”

Comment 6:

How was the severity of itching (mild/moderate/severe) assessed? Did you use any scale? You have also evaluated pruritus (2 items) as a part of the CU-Q2oL tool. Was it done differently?

Response:

The methodology has been clarified:

“Current itching severity was assessed with the following levels: mild (present but not annoying or troublesome), moderate (troublesome but does not interfere with normal daily activity or sleep), and severe (severe itch, which is sufficiently troublesome to interfere with normal daily activity or sleep). These levels were consistent with the Weekly Urticaria Activity Score (UAS7) scale .”

Regarding the CU-Q2oL pruritus scale, it evaluates the impact of pruritus over the past 14 days by asking how much patients were troubled by specific symptoms. This scale focuses on the overall impact of pruritus rather than its severity, so there is no overlap between the UAS7 severity assessment and the CU-Q2oL pruritus items.

Results:

Comment 7:

Though mentioned 358, the sum of n in Table 2 is actually 357. If there was a missing value, please mention.

Response:

We have update the results in Table 2 that fit the sample size 358.

Comment 8:

“The ‘≤Mild (0-15)’, ‘Moderate (16-27)’ and ‘Severe (28-42)’ groups had…” It looks like you combined ‘very mild’ and ‘mild’ together and considered those as ‘mild’. Please add a sentence in the method section addressing this issue.

Response:

We have added the following explanation in the methodology:

“Due to the small number of patients in the ‘very mild’ category (n=3), we combined the ‘very mild’ and ‘mild’ categories into a single group (‘≤Mild [0-15]’) for statistical analysis.”

Conclusion:

Comment 9:

You may consider replacing the word ‘importance’ with ‘necessity’ in the sentence: “The findings underscore the importance of comprehensive and culturally appropriate management strategies to address CSU's physical and psychological impacts.”

Response:

The sentence has been revised:

Original: “The findings underscore the importance of comprehensive and culturally appropriate management strategies…”

Revised: “The findings underscore the necessity of comprehensive and culturally appropriate management strategies…”

Reviewer #4

Comment 1:

While Table 1 provides baseline characteristics, it would be beneficial to stratify these into the three UAS7 severity groups and assess for any statistically significant differences. Even if no statistically significant differences are identified, any apparent variation between groups warrants additional discussion regarding potential for confounding bias.

Response:

We have stratified the baseline characteristics into UAS7 severity groups (Very mild and Mild, Moderate, Severe) in Table 1 and assessed for statistically significant differences. A summary of these findings has been added to the Results section.

We believe these revisions have enhanced the manuscript and addressed all reviewer concerns. Thank you for your thoughtful feedback and the opportunity to improve our work.

Sincerely,

Authors

---

## [Decision Letter · Decision Letter 2]

31 Dec 2024

Factors Associated with Quality of Life of Chronic Spontaneous Urticaria Patients in a Vietnamese Population

PONE-D-24-32836R2

Dear Dr. Le,

We’re pleased to inform you that your manuscript has been judged scientifically suitable for publication and will be formally accepted for publication once it meets all outstanding technical requirements.

Kind regards,

Bharat Bhushan Sharma, M.D.

Academic Editor

PLOS ONE

Additional Editor Comments (optional):

The peer review of the paper has been completed, and it is now suitable for publication in the journal.

Reviewers' comments:

Reviewer's Responses to Questions

**Comments to the Author**

1. If the authors have adequately addressed your comments raised in a previous round of review and you feel that this manuscript is now acceptable for publication, you may indicate that here to bypass the “Comments to the Author” section, enter your conflict of interest statement in the “Confidential to Editor” section, and submit your "Accept" recommendation.

Reviewer #3: All comments have been addressed

Reviewer #4: All comments have been addressed

2. Is the manuscript technically sound, and do the data support the conclusions?

Reviewer #3: Yes

Reviewer #4: Yes

3. Has the statistical analysis been performed appropriately and rigorously? 

Reviewer #3: Yes

Reviewer #4: Yes

4. Have the authors made all data underlying the findings in their manuscript fully available?

Reviewer #3: No

Reviewer #4: Yes

5. Is the manuscript presented in an intelligible fashion and written in standard English?

Reviewer #3: Yes

Reviewer #4: Yes

6. Review Comments to the Author

Reviewer #3: Thank you for addressing the comments. This is an interesting study and after improving, the manuscript looks better.

Reviewer #4: The authors have appropriately responded to the reviewers' comments, including the potential selection bias of three UAS7 severity groups.

7. PLOS authors have the option to publish the peer review history of their article (what does this mean?). If published, this will include your full peer review and any attached files.

Reviewer #3: **Yes: **Ruhul Amin

Reviewer #4: No

---

## [Editor Report · Acceptance letter]

5 Jan 2025

PONE-D-24-32836R2 

PLOS ONE

Dear Dr. Le, 

I'm pleased to inform you that your manuscript has been deemed suitable for publication in PLOS ONE. Congratulations! Your manuscript is now being handed over to our production team.

Kind regards, 

on behalf of

Professor Bharat Bhushan Sharma 

Academic Editor

PLOS ONE
